# In-Context Symmetries: Self-Supervised Learning through Contextual World Models

**Sharut Gupta** [* 1]  **Chenyu Wang** [* 1]  **Yifei Wang** [* 1]  **Tommi Jaakkola** [1]  **Stefanie Jegelka** [1 2]

## Abstract

At the core of self-supervised learning for vision is the idea of learning invariant or equivariant representations with respect to a set of data transformations. This approach, however, introduces strong inductive biases, which can render the representations fragile in downstream tasks that do not conform to these symmetries. In this work, drawing insights from world models, we propose to instead learn a general representation that can adapt to be invariant or equivariant to different transformations by paying attention to *context* — a memory module that tracks task-specific states, actions, and future states. Here, the action is the transformation, while the current and future states respectively represent the input's representation before and after the transformation. Our proposed algorithm, Contextual Self-Supervised Learning (CONTEXTSSL), learns equivariance to all transformations (as opposed to invariance). In this way, the model can learn to encode all relevant features as general representations while having the versatility to tail down to task-wise symmetries when given a few examples as the context. Empirically, we demonstrate significant performance gains over existing methods on equivariance-related tasks, supported by both qualitative and quantitative evaluations.

## 1. Introduction

Self-supervised learning (SSL) of image representations has made remarkable progress in recent years (Chen et al., 2020a; Bardes et al., 2022; Zhou et al., 2022a; Larsson et al., 2016; Gidaris et al., 2018; Bachman et al., 2019; Gidaris et al., 2021; Grill et al., 2020; Shwartz-Ziv et al., 2022; Misra and Maaten, 2020; Chen et al., 2020b; He et al., 2020; Chen and He, 2021; Zbontar et al., 2021; Tomasev et al., 2022; Zhou et al., 2022b), achieving competitive performance to its supervised counterparts on various downstream tasks, such as image classification.

Most of these works are based on the joint-embedding architecture (as shown in Figure 2(a)) which encourages the representations of semantically similar (positive) pairs to be close, and those of dissimilar (negative) pairs to be more orthogonal. Typically, positive pairs are generated by classic data augmentation techniques that correspond to common pretext tasks, e.g., randomizing color, texture, orientation, and cropping. The alignment of representations for positive pairs can be guided by either invariance (Chen et al., 2020a; Bardes et al., 2022; Chen and He, 2021; He et al., 2020; Zbontar et al., 2021; Grill et al., 2020), which promotes insensitivity to these augmentations, or equivariance (Gupta et al., 2023b; Devillers and Lefort, 2023; Dangovski et al., 2022; Garrido et al., 2023b; Assran et al., 2023; Garrido et al., 2024), which maintains sensitivity to them. However, enforcing invariance or equivariance to a pre-defined set of augmentations introduces strong inductive priors which are far from universal across a range of downstream tasks. For example, invariance to image flipping is useful for image classification but can significantly hurt performance on image segmentation, where retaining sensitivity to flipping is crucial. This often results in brittle representations that necessitate retraining the model with different augmentations tailored to each downstream task (Xiao et al., 2021; Dangovski et al., 2022).

This rigidity of traditional SSL methodologies contrasts sharply with human perceptual abilities, which are highly adaptive, tuning into relevant features based on the *context* of the environment or task at hand. For example, humans focus more on color details when identifying flowers, and on spatial orientation such as rotation angle when determining the time on analog clocks. It suggests that the required feature invariances or equivariances should also vary across different tasks or contexts, which motivates our central question.

*Can incorporating context into self-supervised vision algorithms eliminate augmentation-based inductive priors and enable dynamic adaptation to varying task symmetries?*

---

*Equal contribution [1]MIT CSAIL [2]TU Munich. Correspondence to: Sharut Gupta <sharut@mit.edu>.

*Proceedings of the 1st Workshop on In-Context Learning at the 41st International Conference on Machine Learning*, Vienna, Austria. 2024. Copyright 2024 by the author(s).

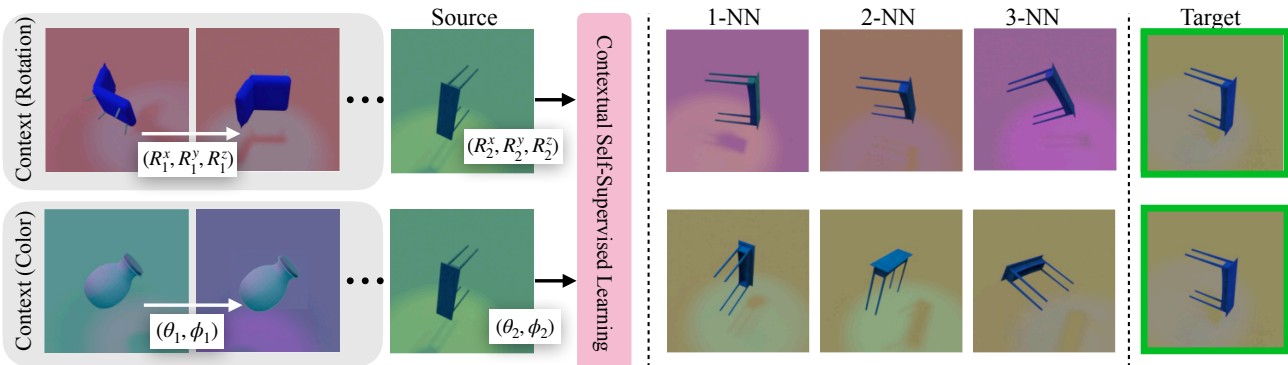

Figure 1: We apply a transformation (rotation or color) on a source image in latent space and retrieve the nearest neighbor (NN) of the predicted representation when the context contains pairs of data transformed by (top row) 3D rotation $(R^x, R^y, R^z)$; (bottom row) color transformation $(\theta, \phi)$. In the top row, we see that CONTEXTSSL learns equivariance to rotation and invariance to color as the NN representations match the target's angle but not its color. In the bottom row, it adapts to the color context and enforces the reverse, be equivariant to color and invariant to rotation.

This work suggests a positive answer to this question by proposing to enhance the current joint embedding architecture with a finite context — an abstract representation of task, containing a few demonstrations that inform about task-specific symmetries, as shown in Figure 2(c). Based on this idea, we propose **Context**ual **S**elf-**S**upervised **L**earning (CONTEXTSSL), a contrastive learning framework that uses a transformer module to adapt to selective invariance or equivariance to transformations by paying attention to context representing a task. Unlike previous approaches with built-in symmetries, the ability of CONTEXTSSL to adapt to varying data symmetries—all without undergoing any parameter updates—enables it to learn a general representation across tasks, devoid of specific inductive priors.

This unique prospect makes our model a promising approach to building world models (Hafner et al., 2020; 2023; Hu et al., 2023; Sekar et al., 2020; Yang et al., 2024) for vision. World models are essential for building representations of the world based on past experiences, akin to how humans form their internal world representations. Recently, efforts have been made to adapt world modeling into vision through Image World Models (IWM) (Garrido et al., 2024) ( Figure 2(b)), that consider transformations as actions and the input and its transformed counterpart as world states at different time steps. However, these approaches also enforce equivariance to a predefined set of actions, such as color jitter. CONTEXTSSL addresses this challenge by enhancing traditional IWMs with context, a model we refer to as *Contextual World Models*. We demonstrate that in the absence of context, CONTEXTSSL learns a general representation by encoding all relevant features and data transformations. As the context increases, the model tailors its symmetries to a task, encouraging equivariance to a subset of transformations and invariance to the rest (as shown in Figure 1).

This approach promotes learning a general representation that can flexibly adapt to the symmetries relevant to various downstream tasks, eliminating the need to learn separate representations for each task. We empirically validate our approach on the 3D Invariant Equivariant Benchmark (3DIEBench) and CIFAR-10, extending to transformations such as rotations, cropping, and blurring.

To summarize, the main contributions of our work are:

- We propose CONTEXTSSL, a self-supervised learning algorithm that adapts to task-specific symmetries by paying attention to context.

- We show that learning with context is prone to identifying shortcuts and subsequently propose two key modules to address it: a context mask and an auxiliary predictor.

- We demonstrate the efficacy of our approach on 3DIEBench and CIFAR10, showing its ability to selectively learn invariance or equivariance to transformations such as color and rotation while maintaining similar performance on invariant (classification) benchmarks. We extend CONTEXTSSL to supervised learning, demonstrating its ability to effectively leverage context to identify features defining a task.

## 2. Augmentation-based Inductive Bias in Self-Supervised Learning

The goal of self-supervised learning (SSL) is to derive meaningful data representations without relying on human-labeled data. Given an unlabeled dataset $\mathcal{D}$, SSL methods learn a representation function $f_\theta : \mathcal{X} \rightarrow \mathcal{Z}$ that maps input data $x \in \mathcal{X}$ to a latent space $\mathcal{Z}$.

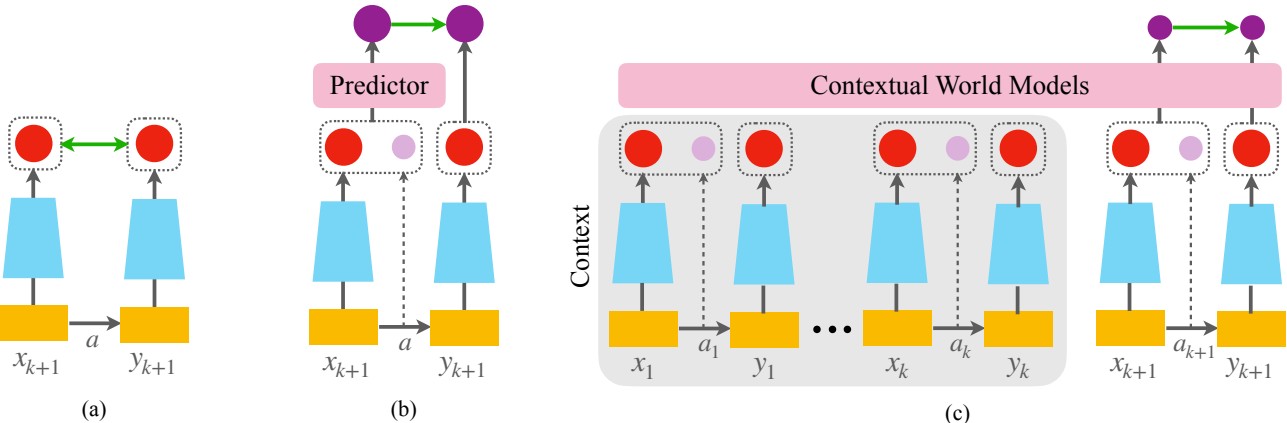

Figure 2: Family of approaches in self-supervised learning (a) **Joint Embedding** methods (Chen et al., 2020a; Bardes et al., 2022; Caron et al., 2021) encode invariances to input transformations $a$ by aligning representations across views of the same image; (b) **Image World Models** (Garrido et al., 2024; Assran et al., 2023) train a world model in the latent space and encode equivariance to input transformations; (c) **Contextual World Models** (ours) selectively enforce equivariance or invariance to a subset of input transformations based on context $\{(x_i, a_i, y_i)\}_{i=1}^{k}$

## 2.1. Role of data augmentations in Self-Supervised Learning

Data augmentations are arguably the most important component in modern SSL methods, where the representation function is learned to map the augmented views of data into latent space. The choice of data augmentations plays a crucial role in the quality of the learned representations. Formally, we define an augmentation $A$ as a random variable distributed over a set of $N$ data transformations with domain $\mathcal{A} = \{a_1, \ldots, a_N\}$, where $a_i : \mathbb{R}^d \to \mathbb{R}^{d'}$ denotes an input mapping, and $d, d'$ are its input and output dimensions, respectively. Among existing SSL methods, there are generally two ways to utilize augmentations, either through invariant learning or equivariant learning. In invariant learning, two random augmentations of the example are drawn and their representations are pulled together during feature learning to be invariant to the data augmentations as shown in Figure 2(a). Instead, in equivariant learning, the features are learned to be sensitive to data augmentations.[1] Formally, for a representation $Z$, one can use $H(A|Z)$ as a measure of the degree of feature invariance or equivariance: if $H(A|Z)$ is relatively small, the representation $Z$ is nearly equivariant to the augmentation $A$; otherwise, if $H(A|Z)$ is very large (close to $H(A)$), $Z$ is invariant to $A$. Recent SSL methods (Gupta et al., 2023b; Garrido et al., 2023b; Park et al., 2022; Devillers and Lefort, 2023; Dangovski et al., 2022) have shown that enforcing equivariance can often lead to

better representations compared to enforcing invariance, for two key reasons: 1) Invariance restricts the expressive power of the features learned as it removes information about features or transformations that may be relevant in fine-grained tasks (Lee et al., 2021; Xie et al., 2022a); 2) contrastive learning benefits from partial invariance through implicit equivariance of the projection head (Jing et al., 2022).

## 2.2. Drawbacks of Hardcoding Symmetries in Self-Supervised Pretraining

As discussed above, a common theme in existing SSL methods is to enforce invariance or equivariance to a specific set of augmentations $A$. For instance, in SimCLR, $A$ is chosen to be a manually selected set of random augmentations such as random cropping, flipping, and color jitter. Therefore, the learned representations, either invariant or equivariant to these augmentations, are tailored to the specific symmetry imposed during pretraining. However, in real world scenarios, there is no single symmetry that is universally applicable across all tasks. For example, object recognition (e.g., a chair) often requires invariance to image color, while certain tasks, e.g., flower recognition, need sensitivity to color information instead. Either to include or not to include color information as part of the augmentations can lead to suboptimal performance in certain tasks, causing a fundamental dilemma in existing SSL. This leads to brittle representations over a range of downstream tasks, as the model needs to be retrained on different augmentations depending on the downstream tasks, as consistently observed in previous works (Xiao et al., 2021; Dangovski et al., 2022).

---

[1]Here, the concept of equivariance is used in a loose sense, meaning that the learned features are sensitive to data augmentations. Note that since some augmentations are non-invertible (e.g., grayscale), they do not form a group, and exact equivariance is not well-defined.

# 3. Beyond Built-in Symmetry: Contextual Self-Supervised Learning

Recognizing the limitations of existing augmentation-specific SSL methods, we propose a new paradigm: **Contextual Self-Supervised Learning** (CONTEXTSSL). Unlike traditional methods, this approach learns a single model that adapts to be either invariant or equivariant based on context-specific augmentations, tailored to the needs of the task or data at hand. Instead of enforcing a fixed set of symmetries, CONTEXTSSL learns these symmetries from contextual cues, thus capturing the unique set of features of downstream tasks. This adaptability allows it to serve as a general-purpose SSL framework, capable of learning from a diverse array of pretraining tasks with varying symmetry priors and seamlessly adapting to different downstream tasks.

To design CONTEXTSSL, we draw inspiration from world modeling (Hafner et al., 2020; 2023; Sekar et al., 2020; Yang et al., 2024), a widely used framework in reinforcement learning (RL). World modeling aims to build representations of the world from past experience by predict the next state $x_{t+1}$ from the current state $x_t$ and action $a_t$. This next state prediction task captures the inherent mechanisms of the system and facilitates decision making. Traditionally applied in RL, the benefits of world modeling in vision have been largely unexplored. Recently, Image World Models (IWM) (Garrido et al., 2023a) established a parallel between world models and the image-based SSL by considering data transformations as actions, the representation of input data as world state at time $t$ and that of the transformed input as next world state. However, IWMs have two key drawbacks: 1) similar to previous SSL approaches, they rely on a predefined set of data augmentations, such as color, which are not tailored to specific downstream tasks and influence the learned features; 2) they lack the memory module of world models that tracks previous experience in terms of past states, actions and corresponding next states and provides context to fully define the current state.

In light of these ideas and challenges, we model CONTEXTSSL in vision self-supervised learning as *Contextual World Models*. In this way, CONTEXTSSL addresses the key drawbacks of IWMs by 1) encouraging the model to preserve all meaningful features to be able to adapt to symmetry from context, and 2) incorporating context to adapt to different task-specific symmetries, removing the need to re-train separate representations for each downstream task. This general ability is akin to human perception that captures versatile aspects of the input, while focusing on specific details depending on the context at hand. For instance, humans focus more on color details when identifying flowers, and on spatial orientation such as rotation angle when determining the time on analog clocks.

## 3.1. Contextual World Models

Drawing inspiration from the in-context learning (Brown et al., 2020) of foundation models in natural language processing, a natural way to incorporate the memory capabilities of world models is by encoding these abilities as contextual information. In this work, we propose an expressive and efficient implementation of CONTEXTSSL through *Contextual World Models*, where we design a transformer-based module to encode the context and extract contextually equivariant or invariant representations. We begin by baking symmetries in the context — $(x, a, y)$ using positive pairs $x$ and $y$ transformed by a series of different augmentations. The key intuition behind our approach is selective inclusion of augmentation parameters for specific transformation groups: excluding parameters enforces invariance, while including them enforces equivariance. This is because providing augmentation parameters allows the model to learn the impact of transformations (equivariance), whereas excluding them during alignment enforces invariance, akin to invariant versus equivariant learning in SSL. We elaborate on these ideas below.

**Symmetries as Context**. Given a set of groups of input transformations $\{\mathcal{G}_1, \ldots, \mathcal{G}_M\}$, the goal of CONTEXTSSL is to build a general representation that is adaptive to a set of multiple symmetries corresponding to these different groups. For example, each data augmentation, e.g., rotation, translation, as well as their compositions, can serve as different transformation groups. Each group $\mathcal{G}_c$ can be represented through the joint distribution $P(x, a, y|\mathcal{G}_c)$, where $x$ is the input sample (sampled from an unlabeled dataset), $a$ represents the parameters of the transformation drawn from $\mathcal{G}_c$ and applied to $x$, and $y$ is the transformed input. In principle, $x$ can be transformed by a composition of augmentations drawn from multiple transformation groups. For instance, in self-supervised learning, it is common to enrich the learning process by transforming an input image through rotations, crops, and blurring. In such a case, $a$ represents a subset of the transformation parameters belonging to the group $\mathcal{G}_c$, applied to $x$ to produce $y$. We approximate this probability distribution by drawing $K$ samples from the joint distribution and form a context $C(\mathcal{G}_c) = [(x_1, a_1, y_1), \ldots, (x_K, a_K, y_K)]$, where $x_i, a_i, y_i \sim P(x, a, y|\mathcal{G}_c), i \in [K]$. Therefore, the goal of ContextSSL is to learn data representations $z = f(x, a|C)$ and $z = f(x|C)$ that are adaptive to the data symmetries informed by the context $C$. Specifically, our goal is to train representations that become more equivariant to the underlying transformation group $\mathcal{G}_c$ with increasing context. Further, if $x$ and $y$ are transformed by augmentations from groups apart from $\mathcal{G}_c$, we aim to learn more invariance to these groups with increase in context $C(\mathcal{G}_c)$. The degree of equivariance of a representation can be quantified by the error in maintaining consistent transformations. Based on

this, a representation $Z$ is considered "more equivariant (invariant)" if it has a lower (higher) error in predicting the transformation parameters i.e. $H(A|Z)$.

**Contextual World Models.** To implement this broad goal, we propose to adaptively learn the symmetries represented by $\mathcal{G}_c$ by training the model:

$$y_i \approx h((x_i, a_i); (x_1, a_1, y_1), \ldots, (x_{i-1}, a_{i-1}, y_{i-1})). \quad (1)$$

While the requested prediction $y_i$ concerns only the inputs $x_i$ and $a_i$, the model can now pay attention to the experience so far, enforcing relevant symmetries for the augmentation group $\mathcal{G}_c$. The predictor $h$ is updated by minimizing the loss at each context length $\sum_{i=1}^{K} \ell(h((x, a_i); C_{i-1}), y_i)$ where $C_i = \{(x_1, a_1, y_1), \ldots, (x_{i-1}, a_{i-1}, y_{i-1})\}$ represents the context before index $i$.

A natural way to facilitate such context-based training is through attention mechanisms in transformer-based autoregressive models. Large language models exhibit a remarkable capability of in-context learning — the ability to generalize to unseen tasks on-the-fly merely by paying attention to a few demonstrative examples of the task. Gupta et al. (2023a) among others, have leveraged this capability to generalize to different distributions merely by paying attention to unlabeled examples from a domain. Inspired by this, we train a decoder-only transformer model in-context by conditioning on the relevant context $C(\mathcal{G}_c)$ representing the transformation group $\mathcal{G}_c$.

### 3.2. Contextual Self-Supervised Learning (CONTEXTSSL)

Motivated by the above ideas, we begin by constructing pairs of points $\{(x_i, y_i)_{i=1}^{K}\}$ by either 1) sampling a transformation group $\mathcal{G}$ and transforming $x_i$ by augmentation from $\mathcal{G}$ to $y_i$; or 2) if available, sampling a meta-latent and its transformation parameters as difference between their individual latent parameters. We use the former construction in datasets such as CIFAR10 but use meta-latents such as 3D pose, lighting etc. for datasets such as 3DIEBench (Garrido et al., 2023b). Note that pairs of data can also be transformed by a series of augmentations sampled from other transformation groups. However, as previously discussed, the transformation parameters used in the context $C(\mathcal{G})$ of group $\mathcal{G}$ are solely those of the augmentations belonging to the group.

Following this, as illustrated in Figure 2, each input sample $\{(x_i, y_i)\}_{i=1}^{K}$ from the context is independently transformed by the encoder into its corresponding latent representation. Next, representations of the input samples $x_i$ are concatenated with their corresponding transformation action $a_i$. This concatenated vector $(x_i, a_i)$ and the representation of the corresponding transformed input $y_i$ collectively form the context corresponding to the symmetry $\mathcal{G}$. The correspond-

ing output embeddings are then aligned using the InfoNCE loss, which is minimized at each context length. If $a_i$ is set to zero for all tokens in a sequence, CONTEXTSSL enforces invariance to $\mathcal{G}$, since it aligns $x_i$ and $y_i$ without conditioning on the transformation parameters. Overall, we optimize the following loss:

$$\mathcal{L}_{\text{algo}}(h) = \mathbb{E}_{\mathcal{G} \sim \{\mathcal{G}_1, \ldots, \mathcal{G}_M\}} \mathbb{E}_{C(\mathcal{G})}$$

$$\sum_{i=1}^{K} \left[ -\log \frac{\exp\left(h((x_i, a_i)|C_i(\mathcal{G}))^{\top} h(y_i|C_i(\mathcal{G}))/\tau\right)}{\sum_{j=1}^{K} \exp\left(h((x_i, a_i)|C_i(\mathcal{G}))^{\top} h(y_j|C_j(\mathcal{G}))/\tau\right)} \right]$$

where transformed data tokens $y_j$ ($j \neq i$) form the negatives. We use a similar symmetric loss term using $y_i$ as the anchor, $(x_i, a_i)$ and $(x_j, a_j)$ ($j \neq i$) as the positive and negatives respectively.

At inference, we tailor the extraction of representations to match the specific requirements of the downstream task, whether it benefits from equivariance or invariance to a transformation group $\mathcal{G}$. In particular, if the task benefits from equivariance, we extract the representations of the test data at the maximum context length used during training $K$, by constructing $\{(x_i, a_i, y_i)\}_{i=1}^{K}$ as its preceding context. Here $a_i$ belongs to the group $\mathcal{G}$ and is used to transform other unlabelled data from the test set $x_i$ into $y_i$. On the contrary, if the downstream task benefits from invariance to the group, we use $\{(x_i, 0, y_i)\}_{i=1}^{K}$ as the preceding context. This notion can be generalized to enforce equivariance to a subset of groups and invariance to another. Specifically, including the augmentation parameters for transformations in a group $\mathcal{G}$ in the context enforces equivariance, while excluding them enforces invariance. In both cases, the data are still transformed using augmentations, regardless of the type of symmetry desired. This flexibility of context creation in CONTEXTSSL allows us to tailor the representations to different symmetries and optimize for the model's performance across a range of tasks. However, this implementation bears two key challenges, as detailed below.

**Context Masking.** Given that $(x_i, a_i)$ precedes $y_i$ in the context sequence, a trivial solution to minimizing the alignment loss arises where the model treats the embeddings of $(x_i, a_i)$ identical to $y_i$ due to its access to $x_i$. This phenomenon, often referred to as shortcut learning, poses a significant challenge as it leads the model to collapse to constant representations for each pair $(x_i, y_i)$, all while perfectly minimizing the loss. We address this challenge by masking out the input token $(x_i, a_i)$ for each token $y_i$ in the context. As a consequence, when encoding the token $y_i$, the transformer only has access to past context

---

[2]In Table 1, both the CONTEXTSSL models are the same and the performance is reported depending on whether the context corresponds to rotation or color augmentation group.

Table 1: Quantitative evaluation of learned representations on invariant (classification) and equivariant (rotation prediction, color prediction) tasks. Additional metrics are reported in Appendix C.6

| $\mathcal{G}$ | Method | Rotation prediction ($R^2$) | | | | | Color prediction ($R^2$) | | | | | Classification (top-1) |
|---|---|---|---|---|---|---|---|---|---|---|---|---|
| | | 0 | 2 | 14 | 30 | 126 | 0 | 2 | 14 | 30 | 126 | Representation |
| | *Invariant* | | | | | | | | | | | |
| | SimCLR | | | 0.506 | | | | | 0.148 | | | **85.3** |
| | SimCLR$^+$(c=0) | | | 0.478 | | | | | 0.070 | | | 83.4 |
| | SimCLR$^+$ | | | 0.247 | | | | | 0.464 | | | 42.3 |
| | VICReg | | | 0.371 | | | | | 0.023 | | | 76.3 |
| | VICReg$^+$(c=0) | | | 0.356 | | | | | 0.062 | | | 73.3 |
| Rotation + Color | *Equivariant* | | | | | | | | | | | |
| | EquiMOD | | | 0.512 | | | | | 0.097 | | | **82.4** |
| | SIE | | | **0.629** | | | | | **0.973** | | | 71.0 |
| | SEN | | | 0.585 | | | | | 0.932 | | | 80.7 |
| Rotation | EquiMOD | | | 0.512 | | | | | 0.097 | | | **82.4** |
| | SIE | | | 0.671 | | | | | **0.011** | | | 77.3 |
| | SEN | | | 0.633 | | | | | 0.055 | | | 81.5 |
| | CONTEXTSSL | 0.734 | 0.740 | 0.743 | 0.743 | **0.744** | 0.908 | 0.664 | 0.037 | 0.023 | 0.046 | 80.4 |
| Color | EquiMOD | | | 0.429 | | | | | 0.859 | | | **82.1** |
| | SIE | | | 0.304 | | | | | 0.975 | | | 70.3 |
| | SEN | | | 0.386 | | | | | 0.949 | | | 77.6 |
| | CONTEXTSSL[2] | 0.735 | 0.614 | 0.389 | 0.345 | **0.344** | 0.908 | 0.981 | 0.985 | 0.986 | **0.986** | 80.4 |

$C_i = \{(x_1, a_1, y_1), \ldots, (x_{i-1}, a_{i-1}, y_{i-1})\}$, excluding its corresponding positive sample $(x_i, a_i)$.

This masking approach ensures that both the anchor and its corresponding positive share the same context, thus promoting the alignment of positive samples based on semantic relationships rather than mere replication. However, as shown in Figure 3 for $p = 0$, a residual challenge of shortcut learning persists when distinguishing the positives from the negatives. Since the context corresponding to each negative is different from that of the anchor and the positive, the model could employ trivial solutions, such as using the mean of the context vector to differentiate between positives and negatives.

To mitigate this issue, we introduce an additional layer of randomness to our masking strategy. Specifically, for each token in the context vector, we implement random masking with a probability $p$ for tokens preceding it. This ensures that for a given anchor token, both the positive and the negatives have different contexts from the anchor, thereby necessitating a deeper, semantic understanding to effectively distinguish the positives from the negatives.

**Avoiding collapse to Invariance.** A trivial but undesirable solution that minimizes our optimization objective is invariance to the input transformations i.e. the trained model can ignore the transformation parameters and collapse back to behaviors associated with invariance-based methods. As illustrated in Figure 4, naively training CONTEXTSSL leads to poor equivariance with respect to the transformations. Previous works (Garrido et al., 2023b) have also identi-

fied this concern and proposed specialized architectures that incorporate transformation parameters directly into the model, thereby outputting the predictor's weights and ensuring effective utilization of these parameters. For our setting, we introduce a rather simple approach that involves jointly training an auxiliary predictor. This predictor is designed to predict the latent transformations of the target sample $y_i$ from the concatenated input vector $(x_i, a_i)$.

## 4. Experimental Results

### 4.1. Quantitative Assessment of Adaptation to Task-Specific Symmetries

We use the 3D Invariant Equivariant Benchmark (3DIEBench) (Garrido et al., 2023b) and CIFAR10 to test our approach. We compare CONTEXTSSL with 1) VICReg (Bardes et al., 2022) and SimCLR (Chen et al., 2020a) among the invariant self-supervised approaches; 2) EquiMOD (Devillers and Lefort, 2023), SEN (Park et al., 2022) and SIE (Garrido et al., 2023b) amongst the equivariant baselines. To discard the performance gains potentially arising from CONTEXTSSL's transformer architecture, for each approach $\mathcal{N}$, we replaces the original projection head or predictor with our transformer model, denoted as $\mathcal{N}^+$. We further test this at For all our equivariant baselines on 3DIEBench, we train equivariant approaches to be equivariant to either only 3D rotation, color transformations, or both. We report the test performance on context lengths 0, 2, 14, 30, and 126. To assess the quality of the invariant representations, we employ linear classification over frozen features.

For the equivariant counterpart, we report $R^2$ on the task of predicting the corresponding transformation. Additionally, we use Mean Reciprocal Rank (MRR) and Hit Rate at $k$ (H@k) to evaluate the performance of our context predictor. More details about pretraining algorithms and training setup are provided in Appendix B.

**Invariant Classification and Equivariant transformation prediction task.** As shown in Table 1, invariant self-supervised learning methods such as SimCLR and VICReg achieve high downstream classification accuracies but underperform in equivariant augmentation prediction tasks. Among the equivariant baselines, EquiMOD persistently maintains its downstream classification accuracy but exhibits improvements in augmentation prediction tasks only when trained to be equivariant to color. SIE and SEN exhibit sensitivity to the trained transformations and remain less sensitive to the others. However, their degree of invariance or equivariance is much worse compared to CONTEXTSSL. In contrast, CONTEXTSSL exhibits equivariance to both rotation and color in the absence of context. As seen from the two rows corresponding to CONTEXTSSL in Table 1, when the context corresponds to pairs of data with transformations sampled from the rotation (color) group, the model adaptively learns to be invariant to color (rotation) while improving equivariance to rotation (color). Appendix C.7 shows that CONTEXTSSL learns equivariance or invariance to the same transformation based on the context.

### 4.2. Role of Context Mask and Auxiliary Predictor

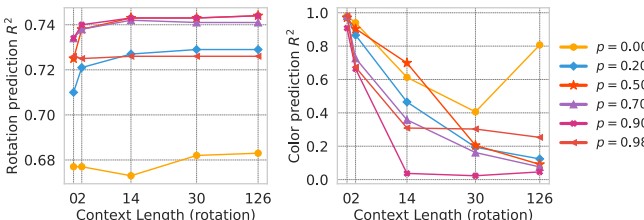

Figure 3: Role of context mask to avoid context based shortcuts in CONTEXTSSL

**Role of Context Mask.** To illustrate how context masking effectively eliminates shortcuts, we conduct an ablation study with varying masking probabilities, detailed in Figure 3. We observed that as masking probability increases, performance on both classification and prediction tasks initially improves but later declines, reaching optimal performance at a masking probability of 90%.

**Role of Auxiliary Predictor.** We demonstrate that the auxiliary predictor is crucial for the model to achieve equivariance. In its absence, as depicted in Figure 4, while the model retains its performance on the invariant classification task, it fails to learn equivariance, and cannot effectively

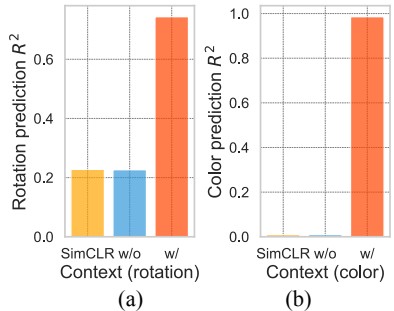

Figure 4: Role of auxiliary predictor to avoid the trivial solution of invariance.

adapt to different contexts.

### 4.3. Qualitative Assessment of Adaptation to Task-Specific Symmetries

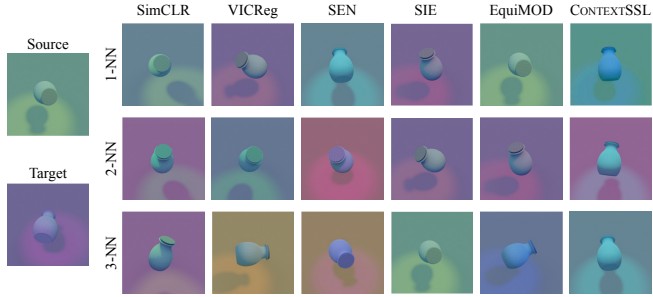

Figure 5: Nearest neighbors of different methods taking as input the source image and rotation angle. CONTEXTSSL aligns best with the rotation angle of the target image

We conduct a qualitative assessment of model performance by taking the nearest neighbors of the predictor output when inputting a source image and a transformation variable, as shown in Figure 5. The nearest neighbors of invariance models (SimCLR and VICReg) have random rotation angles. Equivariance baselines (SEN, SIE, EquiMOD) correctly generate the target rotation angle for some of the 3-nearest neighbors but fail in others. CONTEXTSSL outperforms by successfully identifying the correct angle in all 3-nearest neighbors while remaining invariant to color variations. Additional qualitative assessments for CONTEXTSSL with varying context are provided in Appendix C.3.

### 4.4. Expanding to Diverse Data Transformations

Unlike 3DIEBench where meta-latents for each data are available, we manually construct positives by applying augmentations like crop and blur on CIFAR10. The results for the combinations of crop and blur are reported in Table 2. Consistent with our previous results, while almost retaining the classification performance as SimCLR, CONTEXTSSL

Table 2: Performance of CONTEXTSSL on invariant (classification) and equivariant (crop prediction, blur prediction) tasks in CIFAR-10 under the environment of crop, i.e. CONTEXTSSL (crop), and blur, i.e. CONTEXTSSL (blur).

| Method | Crop prediction ($R^2$) | | | | | Blur prediction ($R^2$) | | | | | Classification (top-1) |
|---|---|---|---|---|---|---|---|---|---|---|---|
| | 0 | 2 | 14 | 30 | 126 | 0 | 2 | 14 | 30 | 126 | Representation |
| SimCLR | | | 0.459 | | | | | 0.371 | | | 89.1 |
| SimCLR$^+$ (c=0) | | | 0.448 | | | | | 0.361 | | | 88.9 |
| SimCLR$^+$ | | | 0.362 | | | | | 0.444 | | | 59.9 |
| CONTEXTSSL (crop) | 0.608 | 0.607 | 0.607 | 0.608 | 0.608 | 0.920 | 0.854 | 0.624 | 0.667 | 0.694 | 88.5 |
| CONTEXTSSL (blur) | 0.609 | 0.482 | 0.434 | 0.417 | 0.465 | 0.920 | 0.923 | 0.925 | 0.925 | 0.925 | 88.5 |

Table 3: Performance of CONTEXTSSL on equivariant tasks (including classificaion) for context-dependent labels. CONTEXTSSL adapts to context-dependent labels with varying context.

| Method | Rotation prediction ($R^2$) | | | | | Color prediction ($R^2$) | | | | | Classification (top-1) | | | | |
|---|---|---|---|---|---|---|---|---|---|---|---|---|---|---|---|
| | 0 | 2 | 14 | 30 | 126 | 0 | 2 | 14 | 30 | 126 | 0 | 2 | 14 | 30 | 126 |
| SimCLR (color) | | | 0.537 | | | | | 0.056 | | | | | 72.0 | | |
| SimCLR (rotation) | | | 0.537 | | | | | 0.056 | | | | | 14.2 | | |
| SimCLR$^+$ (c=0) (color) | | | 0.427 | | | | | -0.007 | | | | | 80.4 | | |
| SimCLR$^+$ (c=0) (rotation) | | | 0.427 | | | | | -0.007 | | | | | 5.2 | | |
| SimCLR$^+$ (color) | | | 0.424 | | | | | 0.243 | | | 16.8 | 15.1 | 15.6 | 14.8 | 14.0 |
| SimCLR$^+$ (rotation) | | | 0.424 | | | | | 0.243 | | | 56.1 | 58.2 | 58.4 | 58.4 | 59.1 |
| CONTEXTSSL (color) | 0.556 | 0.542 | 0.538 | 0.540 | 0.539 | 0.913 | 0.973 | 0.981 | 0.982 | 0.982 | 8.9 | 82.4 | 82.7 | 82.8 | 83.0 |
| CONTEXTSSL (rotation) | 0.556 | 0.624 | 0.661 | 0.665 | 0.666 | 0.913 | 0.379 | 0.111 | 0.095 | 0.093 | 73.5 | 82.7 | 82.6 | 82.6 | 83.0 |

learns to adaptively enforce equivariance to crop (blur) and invariance to blur (crop) depending upon the context. Note that the invariance performance initially improves with increasing context length but then diminishes. This occurs due to the 90% random masking ratio during training, which necessitates out-of-distribution generalization when the context length is large. Results on additional transformation pairs are provided in Appendix C.4.

### 4.5. Context World Models Beyond Self-Supervised Learning

While our analysis has primarily focused on self-supervised learning, the concept of context is versatile and extends beyond representation learning. In principle, irrespective of the task at hand, paying attention to context can learn and identify features defined by it. To validate this, we consider a supervised learning task where our transformer model is trained to directly predict the labels corresponding to an input image. We further corrupt the labels to be directly influenced by the augmentation group transforming the data. Specifically, we add a constant value of 10 to each label if the context corresponds to the rotation group and leave it unchanged otherwise. We report classification performance along with rotation and color prediction equivariant measures. As shown in Table 3, CONTEXTSSL's classification accuracy improves with context, demonstrating its ability to better identify the underlying symmetry group with increase in context. Additional results are provided in Appendix C.5.

Further, CONTEXTSSL serves as a general framework that can adapt to different training regimes

## 5. Conclusion and Future Perspectives

The field of language modeling has witnessed a significant paradigm shift over the past decade, moving towards foundation models that generalize across a variety of tasks either directly or through distillation. However, this shift toward generalization has been conspicuously absent in the vision domain. This is largely because self-supervised approaches for vision still heavily rely on inductive priors strongly introduced by enforcing either invariance or equivariance to data augmentations. This renders representations brittle in downstream tasks that do not conform to these priors and necessitates retraining the representation separately for each task. This work forgoes any notion of pre-defined symmetries and instead trains a model to infer the task-relevant symmetries directly from the context through what we term Contextual Self-Supervised Learning (CONTEXTSSL). The ability of our model to learn selective equivariances and invariances based on mere context opens up new avenues for effectively handling a broader range of tasks, particularly in dynamic environments where the relevance of specific features may change over time.

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

# Appendix

## A. Related Work

**Self-Supervised Learning.** Existing SSL methods generally belong to two categories: invariant learning (Chen et al., 2020a; Bardes et al., 2022; Chen and He, 2021; He et al., 2020; Zbontar et al., 2021; Grill et al., 2020) and equivariant learning. The representative method for invariant learning is contrastive learning, which draws the representations of positive samples together in the latent space such that the representations are invariant to data augmentation. Contrastive learning can learn highly discriminative features at the cost of losing certain image information due to the invariance constraint (Xiao et al., 2021). Motivated by this limitation, recent works explore merging contrastive learning with equivariant learning tasks by separate embedding (Xiao et al., 2021; Garrido et al., 2023b), augmentation-conditioned predictor (Devillers and Lefort, 2023; Garrido et al., 2024), and explicit equivariant transformation (Gupta et al., 2023b). However, existing works still inherit the limitations of contrastive learning: its symmetry prior is built on a given set of manual augmentations and is not adaptive to downstream tasks. In contrast, our method enables the contextual world model to adapt its symmetry to the contextual data, which is more flexible and generalizable to various tasks.

**World Models.** World modeling has achieved notable success in reinforcement learning (RL) for model-based planning (Ha and Schmidhuber, 2018; Sekar et al., 2020; Hafner et al., 2020) and vision (Hafner et al., 2023; Hu et al., 2023; Yang et al., 2024), where it involves predicting future states based on current observations and actions. This concept, however, has not yet been fully leveraged in visual representation learning. Nevertheless, Garrido et al. (2024) shows that several families of self-supervised learning approaches can be reformulated through the lens of world modeling. Equivariant self-supervised learning methods. Specifically, Masked Image Modeling approaches (He et al., 2022; Bao et al., 2022; El-Nouby et al., 2024; Xie et al., 2022b) consider masked pixels and target pixel reconstruction as their action and next state. Other equivariant learning approaches (Devillers and Lefort, 2023; Park et al., 2022; Garrido et al., 2023b) consider data transformations and representation of the target image as their action and next state pair. However, unlike true world modeling, these approaches do not track past experiences, a component critical for generalization. Our method instead leverages context to track past experiences in terms of state, action, and next-state triplets, enabling it to adapt and generalize to varying environments.

**In-context Learning.** Our work is inspired by and extends the concept of in-context learning (ICL) (Brown et al., 2020) to training. Initially studied in the context of language, in-context learning has recently been adapted for vision tasks (Gupta et al., 2023a; Wang et al., 2023; Bar et al., 2022; Li and Liang, 2021), allowing models to infer environmental features or tasks directly from input prompts without predefined notions. For example, Visual Prompting (Wang et al., 2023; Bar et al., 2022) uses a task input/output example pair and a query image at test time, and uses inpainting to generate the desired output. Gupta et al. (2023a) propose using unlabeled data as context at training to extract environment-specific signals and address domain generalization. ICL has been extensively explored in various domains, including vision, language, and multimodal tasks. However, our work is the first to apply ICL to vision self-supervised representation learning.

## B. Supplementary experimental details and assets disclosure

To evaluate the efficacy of our proposed algorithm CON-TEXTSSL, our experiments are designed to address the following questions:

1. How does CONTEXTSSL fare against competitive invariant and equivariant self-supervised learning approaches in terms of performance across varying context sizes and different sets of data transformations?

2. How effectively can CONTEXTSSL identify task-specific symmetries, both within the scope of self-supervised learning and beyond?

3. What roles do specific components such as selective masking and the auxiliary latent transformation predictor play in facilitating the learning of general and context-adaptable representations?

### B.1. Assets

We do not introduce new data in the course of this work. Instead, we use publicly available widely used image datasets for the purposes of benchmarking and comparison.

### B.2. Hardware and setup

Each experiment was conducted on 1 NVIDIA Tesla V100 GPUs, each with 32GB of accelerator RAM. The CPUs used were Intel Xeon E5-2698 v4 processors with 20 cores and 384GB of RAM. All experiments were implemented using the PyTorch deep learning framework.

### B.3. Datasets

**3D Invariant Equivariant Benchmark (3DIEBench).** To test equivariance and invariance to multiple data transformations, we use the 3D Invariant Equivariant

Benchmark (3DIEBench) (Garrido et al., 2023b) which has been specifically designed to address the limitations of existing datasets in evaluating invariant and equivariant representations. It contains images of 3D objects along with their latent parameters such as object rotation, lighting color, and floor color. Since we have access available to individual meta latent parameters, transformation parameters between two views of an object are calculated as the difference between their individual latents. We test our approach on 3DIEBench under two settings 1) Considering two transformation groups: rotation and color with the aim of learning invariance to one and equivariance to another after conditioning on context; 2) Considering one transformation group, say rotation and learning to enforce invariance or equivariance to rotation with context. As previously mentioned, all methods are trained for 1000 epochs using a batch size of 512 on 128×128 resolution images. We use the standard training, validation and test splits, made publicly available by the authors (Garrido et al., 2023b).

**CIFAR10.** 3DIEBench dataset is limited to only rotations and color as transformation groups. We extend our approach to include more common self-supervised benchmarks, such as CIFAR-10, incorporating transformations like blurring, color jitter, and cropping. Unlike 3DIEBench, we manually construct positive pairs by applying compositions of these handcrafted augmentations. We consider three transformation groups: crop, blur and color. Similar to 3DIEBench, we consider combinations of two groups for each training run. We use the standard training, validation and test splits.

**B.4. Baseline Algorithms**

Among the invariant self-supervised approached, we compare our approach to VICReg (Bardes et al., 2022) and and SimCLR (Chen et al., 2020a). For each method, comparisons are drawn using their originally proposed architectures. For the equivariant baselines, we consider EquiMOD (Devillers and Lefort, 2023), SIE (Garrido et al., 2023b) and SEN (Park et al., 2022). Similar to Garrido et al. (2023b), For SEN, we use the InfoNCE loss instead the original triplet loss. To discard the performance gains potentially arising from CONTEXTSSL's transformer architecture, for each approach, we consider an additional baseline that replaces the original projection heads or predictor with our transformer model. Given an algorithm name $\mathcal{N}$, we refer to this baseline as $\mathcal{N}^{+}$. Amongst these, we report the best performing variant in our results. For $\mathcal{N}^{+}$, we conduct analysis in two distinct settings: 1) a 'no context' or $c = 0$ invariant condition, and 2) a fully contextualized setting with a context length of 126.

**B.5. Training Protocol**

To ensure a fair comparison across different algorithms for each dataset, we use a standardized neural network backbone. Precisely, for our encoder, we use a ResNet-18 backbone pre-trained on ImageNet. For CONTEXTSSL, output features from the encoder are transformed into the context sequence, which is then processed by the decoder-only Transformer (Vaswani et al., 2017) from the GPT-2 Transformer family (Radford et al., 2019). Our model configuration includes 3 layers, 4 attention heads, and a 2048-dimensional embedding space, consistently applied across all datasets. Linear layers are utilized to convert the input sequence into the transformer's latent embedding of dimension 2048 and to map the predicted output vectors to the output space of dimension 512.

We fix the maximum training context length to 128. Since for every $y$, the corresponding token $(x_i, a_i)$ is masked out, context length $L$ corresponds to effective context length $L - 2$. Thus, we report CONTEXTSSL's performance over varying test context length of 0, 2, 14, 30 and 126. On all datasets, we train CONTEXTSSL with the Adam optimizer with a learning rate of $5e^{-5}$ and weight decay $1e-3$. For baseline self-supervised approaches, in their original architecture, we use a learning rate of $1e^{-3}$ with no weight decay. However, when tested using the transformer architecture, we choose one of the above two optimizer hyperpameters. Consequently, performance of the best performing model is reported among the two baselines. Similar to Garrido et al. (2023b), we report hyper-parameters and architectures specific to each method:

- **SimCLR (Chen et al., 2020a)** We train using a 2048-2048-2048 dimensional multi-layered perceptron (MLP) based projection head with a temperature of 0.5.

- **VICReg (Bardes et al., 2022)** We train using a 2048-2048-2048 MLP for the projection head and use weight of 10 for both the invariance loss and variance loss and 1 for covariance loss.

- **SEN (Park et al., 2022)** Similar to other approaches we use a projection head of dimension 2048-2048-2048 and temperature 0.1.

- **EquiMod (Devillers and Lefort, 2023)** We use the standatd projection head of dimensions 1024-1024-128 and use equal weighing of the invariance and the equivariance loss.

- **SIE** (Garrido et al., 2023b) We use two 1024-1024-1024 projection heads, one for invariant latent space and other for equivariant. When trained to learn equivariance to only rotation or only color, we use weight

of 10 for both the invariance loss and variance loss, 1 for the covariance loss and 4.5 for the equivariant loss. However, when trained to be equivariant to both rotation and color jointly, we use 10 as the equivariant weight.

## B.6. Evaluation metrics

In line with established self-supervised learning methodologies, we begin by assessing the quality of the learned representations through downstream tasks. For evaluating invariant representations, we employ linear classification over frozen features. To evaluate equivariant representations, we predict the corresponding data transformation. This prediction takes representations from two differently transformed views of the same object and regresses on the applied transformation between them. Further, we use Mean Reciprocal Rank (MRR) and Hit Rate at $k$ (H@k) to evaluate the performance for our context predictor. Given the source data and the transformation action, we identify the $k$ nearest neighbors in the embedding space. MRR is calculated as the average reciprocal rank of the target embedding within these nearest neighbors. Hit rate-k (H@k) assigns a score of 1 if the target embedding is within the k-nearest neighbors of the predicted embedding and 0 otherwise. Similar to Garrido et al. (2023b), we restrict the search for nearest neighbors to different views of the same object, thus ensuring that the predictor is not penalized for retrieving an incorrect object in a pose similar to the correct one.

## C. Additional Experiments

### C.1. Quantitative Assessment of Adaptation to Task-Specific Symmetries

In this section, we present additional results on the quantitative assessment of model performance on 3DIEBench, including the evaluation of learned representations on equivariant tasks (rotation and color prediction) to predict individual latent values. In contrast, the results in Table 1 focus on predicting relative latent values between pairs of image embeddings as inputs.

#### C.1.1. INVARIANT CLASSIFICATION AND EQUIVARIANT TRANSFORMATION PREDICTION TASK

As shown in Table 4, invariant self-supervised learning methods such as SimCLR and VICReg underperform in equivariant augmentation prediction tasks. The equivariant baselines, EquiMOD, SIE, and SEN, exhibit improvements compared to the invariant baselines in some of the augmentation prediction tasks. However, their degree of equivariance is much worse compared to CONTEXTSSL. Besides, aligning them with different targeted symmetry groups requires retraining the entire model. In contrast, CONTEXTSSL employs a single model capable of learning equivariance to rotation and invariance to color (or vice versa) based on the given context. As seen from the two rows corresponding to CONTEXTSSL Table 1, when the context corresponds to pairs of data with transformations sampled from the rotation (color) group, the model adaptively learns to be invariant to color (rotation) while retaining equivariance to rotation (color).

Results in Table 1 are the average value over three random seeds. We provide the standard deviation for rotation and color prediction of CONTEXTSSL in Table 5 and Table 6.

#### C.1.2. EQUIVARIANT MEASURES BASED ON NEAREST NEIGHBOURS RETRIEVAL

Similar to **??**, we provide the performance of CONTEXTSSL on MRR and H@k compared to baseline methods with trained equivariance to rotation. While **??** uses the validation set data as the retrieval library, Table 7 provides the results using the training set data. CONTEXTSSL outperforms the baseline models, and its performance on all the metrics consistently improves with increasing context length, showing adaptation to rotation-specific features.

### C.2. Role of Context Mask and Auxiliary Predictor

In this section, we provide additional results for the role of context mask and auxiliary predictor.

#### C.2.1. ROLE OF CONTEXT MASK

In addition to Figure 3, we provide the performance of the rotation and color prediction tasks with varying masking probabilities under the environment of color in Figure 7. We observed that as masking probability increases, performance on both classification and prediction tasks initially improves but later declines, reaching optimal performance at a masking probability of 90%.

Results in Figure 3 and Figure 7 are the average value over three random seeds. We provide the standard deviation for rotation and color prediction of CONTEXTSSL in Table 8 and Table 9.

#### C.2.2. ROLE OF AUXILIARY PREDICTOR

We provide the complete results corresponding to Figure 4 in Table 10 to demonstrate that the auxiliary predictor is crucial for the model to achieve equivariance. In its absence, while the model retains its performance on the invariant classification task, it fails to learn equivariance, performs similarly to the invariant models, and cannot effectively adapt to different contexts.

Table 4: Quantitative evaluation of learned representations on equivariant (rotation prediction, color prediction) tasks to predict individual latent values.

| $\mathcal{G}$ | Method | Rotation prediction ($R^2$) | | | | | Color prediction ($R^2$) | | | | |
|---|---|---|---|---|---|---|---|---|---|---|---|
| | | 0 | 2 | 14 | 30 | 126 | 0 | 2 | 14 | 30 | 126 |
| | *Invariant* | | | | | | | | | | |
| | SimCLR | | | 0.791 | | | | | 0.137 | | |
| | SimCLR$^+$(c=0) | | | 0.773 | | | | | 0.061 | | |
| | SimCLR$^+$ | | | 0.544 | | | | | 0.498 | | |
| | VICReg | | | 0.660 | | | | | 0.011 | | |
| | VICReg$^+$(c=0) | | | 0.615 | | | | | 0.061 | | |
| Rotation + Color | *Equivariant* | | | | | | | | | | |
| | EquiMOD | | | 0.712 | | | | | 0.221 | | |
| | SIE | | | **0.760** | | | | | **0.972** | | |
| | SEN | | | 0.617 | | | | | 0.888 | | |
| Rotation | EquiMOD | | | 0.707 | | | | | 0.033 | | |
| | SIE | | | 0.790 | | | | | **0.001** | | |
| | SEN | | | 0.723 | | | | | 0.437 | | |
| | CONTEXTSSL[3] | 0.838 | 0.839 | 0.840 | 0.840 | **0.840** | 0.895 | 0.620 | 0.021 | 0.014 | 0.021 |
| Color | EquiMOD | | | 0.660 | | | | | 0.855 | | |
| | SIE | | | **0.560** | | | | | 0.974 | | |
| | SEN | | | 0.713 | | | | | 0.876 | | |
| | CONTEXTSSL[4] | 0.838 | 0.800 | 0.699 | 0.666 | 0.685 | 0.895 | 0.981 | 0.985 | 0.985 | **0.986** |

Table 5: Performance of CONTEXTSSL in 3DIEBench in rotation prediction under the environment of rotation, i.e. CONTEXTSSL (rotation), and color, i.e. CONTEXTSSL (color), with standard deviations over three random seeds.

| Method | Rotation prediction ($R^2$) | | | | |
|---|---|---|---|---|---|
| | 0 | 2 | 14 | 30 | 126 |
| CONTEXTSSL (rotation) | $0.734 \pm 0.002$ | $0.740 \pm 0.004$ | $0.743 \pm 0.001$ | $0.743 \pm 0.001$ | $0.744 \pm 0.001$ |
| CONTEXTSSL (color) | $0.735 \pm 0.001$ | $0.614 \pm 0.108$ | $0.389 \pm 0.054$ | $0.345 \pm 0.040$ | $0.344 \pm 0.003$ |

Table 6: Performance of CONTEXTSSL in 3DIEBench in color prediction under the environment of rotation, i.e. CONTEXTSSL (rotation), and color, i.e. CONTEXTSSL (color), with standard deviations over three random seeds.

| Method | Color prediction ($R^2$) | | | | |
|---|---|---|---|---|---|
| | 0 | 2 | 14 | 30 | 126 |
| CONTEXTSSL (rotation) | $0.908 \pm 0.002$ | $0.664 \pm 0.166$ | $0.037 \pm 0.010$ | $0.023 \pm 0.001$ | $0.046 \pm 0.007$ |
| CONTEXTSSL (color) | $0.908 \pm 0.002$ | $0.981 \pm 0.002$ | $0.985 \pm 0.001$ | $0.986 \pm 0.001$ | $0.986 \pm 0.001$ |

## C.3. Qualitative Assessment of Adaptation to Task-Specific Symmetries

### C.3.1. COMPARISON WITH BASELINE APPROACHES

We provide additional results to the qualitative assessment comparing with different models in Figure 8. The nearest neighbors of invariance models (SimCLR and VICReg) have random rotation angles. Equivariance baselines (SEN, SIE, EquiMOD) correctly generate the target rotation angle for some of the 3-nearest neighbors but fail in others. CONTEXTSSL outperforms by successfully identifying the correct angle in all 3-nearest neighbors while remaining invariant to color variations.

Table 7: Quantitative evaluation of learned predictors equivariant to only rotation based on Mean Reciprocal Rank (MRR) and Hit Rate H@k on training dataset. CONTEXTSSL learns to be more equivariant to rotation with context.

| Method | MRR (↑) | | | | | H@1 (↑) | | | | | H@5 (↑) | | | | |
|---|---|---|---|---|---|---|---|---|---|---|---|---|---|---|---|
| | 0 | 2 | 14 | 30 | 126 | 0 | 2 | 14 | 30 | 126 | 0 | 2 | 14 | 30 | 126 |
| EquiMOD | | | 0.17 | | | | | 0.06 | | | | | 0.24 | | |
| SEN | | | 0.17 | | | | | 0.06 | | | | | 0.24 | | |
| CONTEXTSSL | 0.282 | 0.321 | 0.470 | 0.498 | **0.531** | 0.132 | 0.263 | 0.375 | 0.398 | **0.402** | 0.436 | 0.495 | 0.650 | 0.669 | **0.680** |

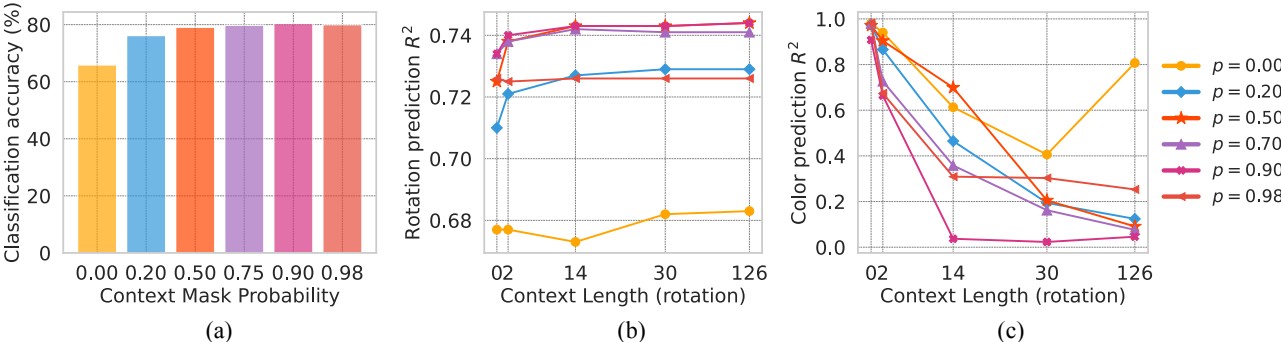

Figure 6: Role of context mask to avoid context based shortcuts in CONTEXTSSL under *rotation* context

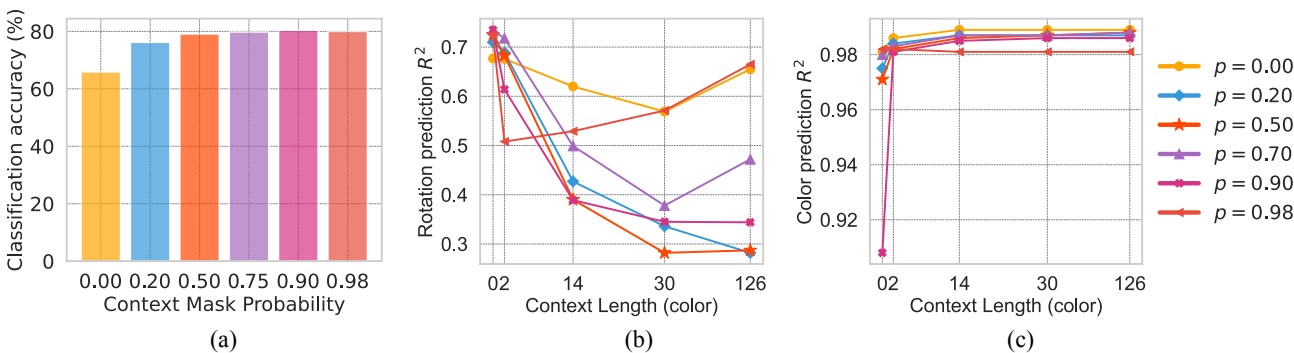

Figure 7: Role of context mask to avoid context based shortcuts in CONTEXTSSL under *color* context

### C.3.2. NEAREST NEIGHBOUR RETRIEVAL WITH VARYING CONTEXT

In this section, we conduct a qualitative assessment of model performance by taking the nearest neighbors of the predictor output when inputting a source image and a transformation variable, and show the change in retrieving quality in Figure 9, Figure 10, and Figure 11. We observe that the nearest neighbors have a closer rotation angle (color) to the target image under rotation (color) context as context length increases, indicating CONTEXTSSL's ability to adapt to the given context as context length increases.

### C.4. Expanding to Diverse Data Transformations

Unlike 3DIEBench where meta-latents for each data are available, we manually construct positives by applying aug-

mentations like crop and blur on CIFAR10. The results for the combinations of crop and blur are reported in Table 2. We additionally provide the results for the combinations of crop and color in Table 12 and crop and blur in Table 2. Consistent with our previous results, while almost retaining the classification performance as SimCLR, CONTEXTSSL learns to adaptively enforce equivariance and invariance to different environments depending upon the context.

In addition to the results for predicting relative latent values between pairs of image embeddings as input in Table 2, Table 12, and Table 11, we provide the evaluation of learned representations on equivariant tasks (rotation and color prediction) to predict individual latent values, as shown in Table 13, Table 15, and Table 14 respectively. Both results lead to the same conclusion, that CONTEXTSSL is able to adaptively enforce equivariance and invariance to different

Table 8: Performance of CONTEXTSSL rotation prediction tasks in 3DIEBench under different random masking probabilities, with standard deviations over three random seeds.

| Context | Probability | Rotation prediction ($R^2$) | | | | |
|---|---|---|---|---|---|---|
| | | 0 | 2 | 14 | 30 | 126 |
| Rotation | 0.00 | $0.677 \pm 0.004$ | $0.677 \pm 0.002$ | $0.673 \pm 0.009$ | $0.682 \pm 0.003$ | $0.683 \pm 0.003$ |
| | 0.20 | $0.710 \pm 0.002$ | $0.721 \pm 0.006$ | $0.727 \pm 0.002$ | $0.729 \pm 0.001$ | $0.729 \pm 0.001$ |
| | 0.50 | $0.725 \pm 0.001$ | $0.738 \pm 0.005$ | $\mathbf{0.743 \pm 0.001}$ | $\mathbf{0.743 \pm 0.001}$ | $\mathbf{0.744 \pm 0.001}$ |
| | 0.75 | $\mathbf{0.734 \pm 0.002}$ | $0.738 \pm 0.006$ | $0.742 \pm 0.004$ | $0.741 \pm 0.004$ | $0.741 \pm 0.002$ |
| | 0.90 | $\mathbf{0.734 \pm 0.002}$ | $\mathbf{0.740 \pm 0.004}$ | $\mathbf{0.743 \pm 0.001}$ | $\mathbf{0.743 \pm 0.001}$ | $\mathbf{0.744 \pm 0.001}$ |
| | 0.98 | $0.726 \pm 0.002$ | $0.725 \pm 0.003$ | $0.726 \pm 0.002$ | $0.726 \pm 0.003$ | $0.726 \pm 0.003$ |
| Color | 0.00 | $\mathbf{0.677 \pm 0.004}$ | $0.676 \pm 0.005$ | $0.620 \pm 0.019$ | $0.569 \pm 0.019$ | $0.655 \pm 0.010$ |
| | 0.20 | $0.710 \pm 0.002$ | $0.689 \pm 0.013$ | $0.427 \pm 0.031$ | $0.336 \pm 0.007$ | $\mathbf{0.282 \pm 0.022}$ |
| | 0.50 | $0.725 \pm 0.001$ | $0.683 \pm 0.006$ | $0.390 \pm 0.031$ | $\mathbf{0.282 \pm 0.013}$ | $0.287 \pm 0.002$ |
| | 0.75 | $0.734 \pm 0.002$ | $0.718 \pm 0.002$ | $0.499 \pm 0.035$ | $0.378 \pm 0.054$ | $0.472 \pm 0.015$ |
| | 0.90 | $0.735 \pm 0.001$ | $0.614 \pm 0.108$ | $\mathbf{0.389 \pm 0.054}$ | $0.345 \pm 0.040$ | $0.344 \pm 0.003$ |
| | 0.98 | $0.726 \pm 0.002$ | $\mathbf{0.508 \pm 0.127}$ | $0.529 \pm 0.141$ | $0.571 \pm 0.125$ | $0.665 \pm 0.023$ |

Table 9: Performance of CONTEXTSSL color prediction tasks in 3DIEBench under different random masking probabilities, with standard deviations over three random seeds.

| Context | Probability | Color prediction ($R^2$) | | | | |
|---|---|---|---|---|---|---|
| | | 0 | 2 | 14 | 30 | 126 |
| Rotation | 0.00 | $0.981 \pm 0.002$ | $0.940 \pm 0.033$ | $0.613 \pm 0.123$ | $0.406 \pm 0.125$ | $0.807 \pm 0.080$ |
| | 0.20 | $0.975 \pm 0.001$ | $0.866 \pm 0.171$ | $0.465 \pm 0.113$ | $0.194 \pm 0.057$ | $0.124 \pm 0.027$ |
| | 0.50 | $0.971 \pm 0.002$ | $0.904 \pm 0.086$ | $0.699 \pm 0.028$ | $0.205 \pm 0.054$ | $0.091 \pm 0.016$ |
| | 0.75 | $0.980 \pm 0.001$ | $0.727 \pm 0.351$ | $0.358 \pm 0.233$ | $0.162 \pm 0.021$ | $0.076 \pm 0.009$ |
| | 0.90 | $\mathbf{0.908 \pm 0.002}$ | $\mathbf{0.664 \pm 0.166}$ | $\mathbf{0.037 \pm 0.010}$ | $\mathbf{0.023 \pm 0.001}$ | $\mathbf{0.046 \pm 0.007}$ |
| | 0.98 | $0.982 \pm 0.001$ | $0.674 \pm 0.368$ | $0.309 \pm 0.139$ | $0.303 \pm 0.118$ | $0.253 \pm 0.033$ |
| Color | 0.00 | $0.981 \pm 0.002$ | $\mathbf{0.986 \pm 0.002}$ | $\mathbf{0.989 \pm 0.001}$ | $\mathbf{0.989 \pm 0.001}$ | $\mathbf{0.989 \pm 0.001}$ |
| | 0.20 | $0.975 \pm 0.001$ | $0.984 \pm 0.002$ | $0.987 \pm 0.001$ | $0.987 \pm 0.001$ | $0.987 \pm 0.001$ |
| | 0.50 | $0.971 \pm 0.002$ | $0.982 \pm 0.002$ | $0.986 \pm 0.002$ | $0.987 \pm 0.002$ | $0.988 \pm 0.001$ |
| | 0.75 | $0.980 \pm 0.001$ | $0.983 \pm 0.001$ | $0.987 \pm 0.001$ | $0.987 \pm 0.001$ | $0.988 \pm 0.001$ |
| | 0.90 | $0.908 \pm 0.002$ | $0.981 \pm 0.002$ | $0.985 \pm 0.001$ | $0.986 \pm 0.001$ | $0.986 \pm 0.001$ |
| | 0.98 | $\mathbf{0.982 \pm 0.001}$ | $0.982 \pm 0.001$ | $0.981 \pm 0.001$ | $0.981 \pm 0.001$ | $0.981 \pm 0.001$ |

Table 10: Performance of CONTEXTSSL on classification, rotation and color prediction tasks in 3DIEBench with and without the auxiliary predictor

| Method | Rotation prediction ($R^2$) | | | | | Color prediction ($R^2$) | | | | | Classification (top-1) |
|---|---|---|---|---|---|---|---|---|---|---|---|
| | 0 | 2 | 14 | 30 | 126 | 0 | 2 | 14 | 30 | 126 | Representation |
| SimCLR | | | 0.227 | | | | | -0.004 | | | 85.3 |
| SimCLR$^+$ (c=0) | | | 0.230 | | | | | -0.004 | | | 83.4 |
| SimCLR$^+$ | | | 0.245 | | | | | 0.028 | | | 42.3 |
| CONTEXTSSL (w/o) (rotation) | 0.227 | 0.227 | 0.226 | 0.226 | 0.227 | -0.003 | -0.003 | -0.003 | -0.004 | -0.004 | 80.8 |
| CONTEXTSSL (w/o) (color) | 0.227 | 0.227 | 0.226 | 0.226 | 0.227 | -0.003 | -0.003 | -0.003 | -0.004 | -0.004 | 80.8 |
| CONTEXTSSL (rotation) | 0.734 | 0.740 | 0.743 | 0.743 | 0.744 | 0.908 | 0.664 | 0.037 | 0.023 | 0.046 | 80.4 |
| CONTEXTSSL (color) | 0.735 | 0.614 | 0.389 | 0.345 | 0.344 | 0.908 | 0.981 | 0.985 | 0.986 | 0.986 | 80.4 |

environments depending upon the context.

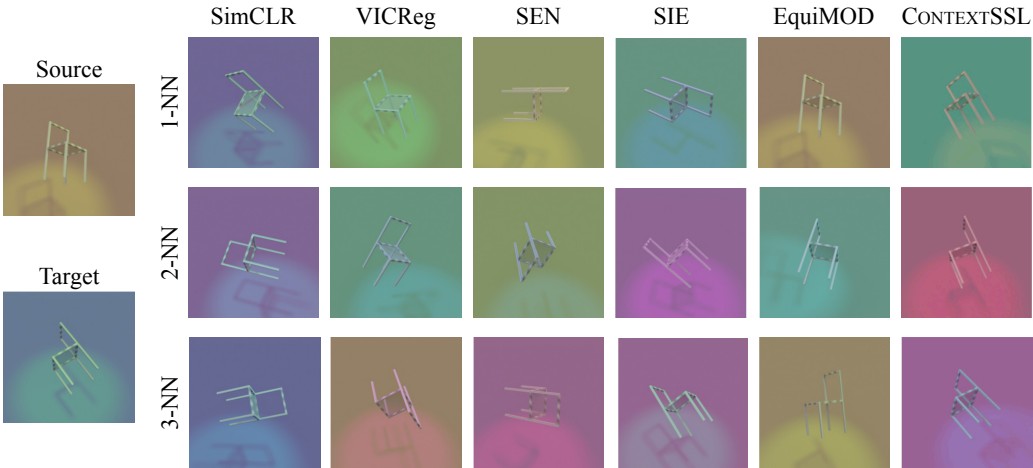

Figure 8: Nearest neighbors of different methods taking as input the source image and rotation angle. CONTEXTSSL aligns best with the rotation angle of the target image.

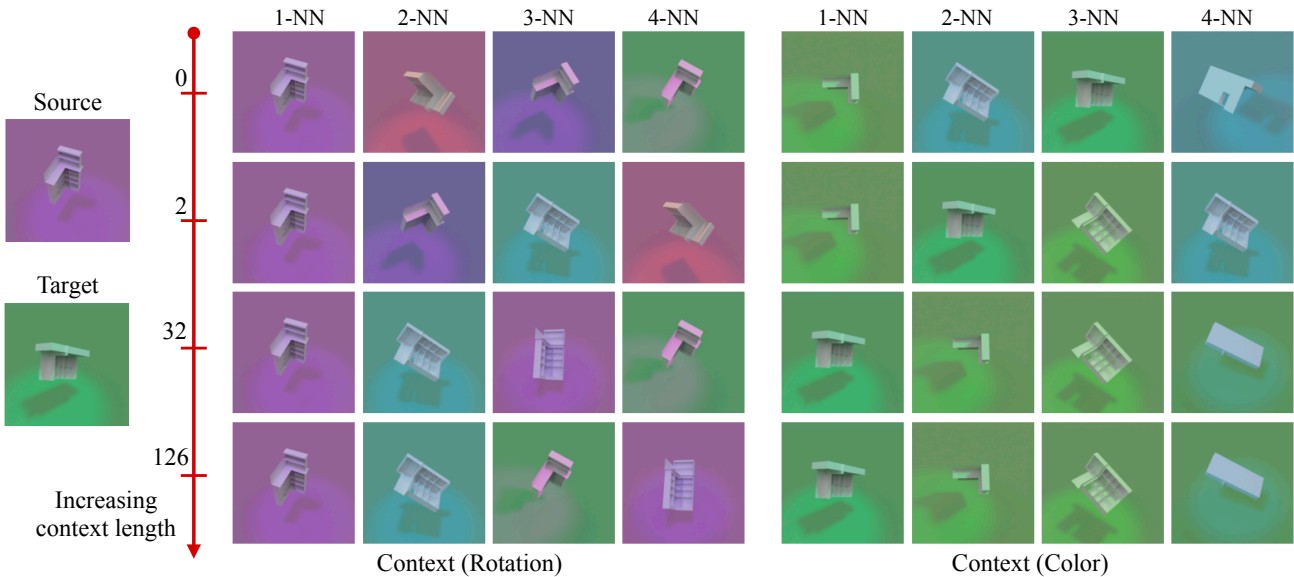

Figure 9: Nearest neighbors of CONTEXTSSL taking as input the source image and rotation angle at different context lengths. As context increases, CONTEXTSSL aligns better with the rotation angle (color) of the target image when the context is based on rotation (color).

Table 11: **CIFAR-10 Color-Blur.** Performance of CONTEXTSSL on invariant (classification) and equivariant (color prediction, blur prediction) tasks in CIFAR-10 under the environment of color, i.e. CONTEXTSSL (color), and blur, i.e. CONTEXTSSL (blur).

| Method | Color prediction ($R^2$) | | | | | Blur prediction ($R^2$) | | | | | Classification (top-1) |
|---|---|---|---|---|---|---|---|---|---|---|---|
| | 0 | 2 | 14 | 30 | 126 | 0 | 2 | 14 | 30 | 126 | Representation |
| SimCLR | | | 0.154 | | | | | 0.371 | | | 89.1 |
| SimCLR$^+$ (c=0) | | | 0.054 | | | | | 0.361 | | | 88.9 |
| SimCLR$^+$ | | | 0.318 | | | | | 0.444 | | | 59.9 |
| CONTEXTSSL (color) | 0.518 | 0.519 | 0.519 | 0.519 | 0.519 | 0.916 | 0.793 | 0.699 | 0.735 | 0.823 | 88.9 |
| CONTEXTSSL (blur) | 0.518 | 0.353 | 0.241 | 0.259 | 0.333 | 0.916 | 0.916 | 0.916 | 0.916 | 0.917 | 88.8 |

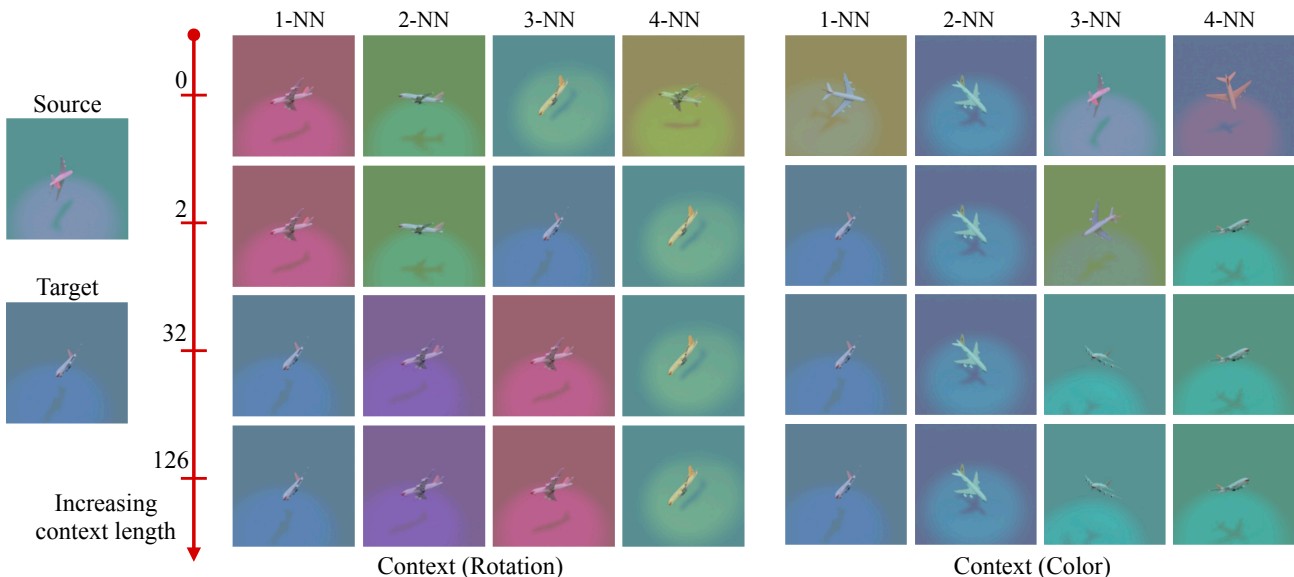

Figure 10: Nearest neighbors of CONTEXTSSL taking as input the source image and rotation angle at different context lengths. As context increases, CONTEXTSSL aligns better with the rotation angle (color) of the target image when the context is based on rotation (color).

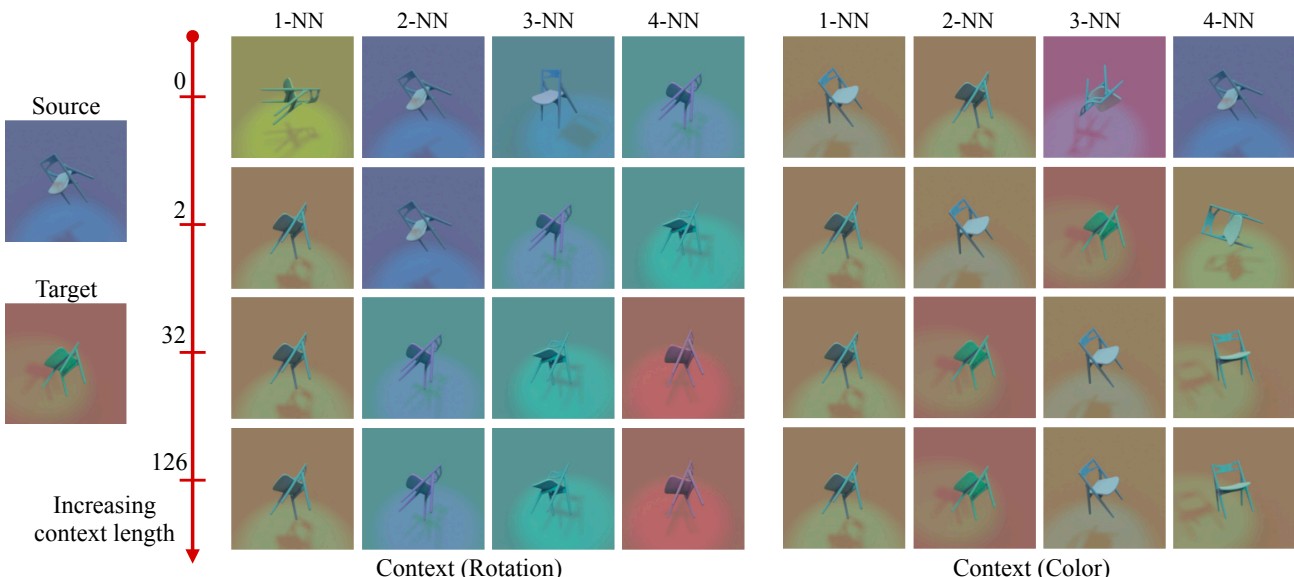

Figure 11: Nearest neighbors of CONTEXTSSL taking as input the source image and rotation angle at different context lengths. As context increases, CONTEXTSSL aligns better with the rotation angle (color) of the target image when the context is based on rotation (color).

## C.5. Context World Models Beyond Self-Supervised Learning

We report classification performance along with rotation and color prediction equivariant measures. The results for predicting relative values are shown in Table 3 and the results for predicting individual latent values are shown in Table 16. The equivariance (invariance) performance of

CONTEXTSSL improves with increased context.

## C.6. Performance on Encoder Representations and Predictor Embedding

We analyze the difference between the performance on representation and the performance on predictor embedding for both the invariance (classification) task and equivariance

Table 12: **CIFAR-10 Crop-Color.** Performance of CONTEXTSSL on invariant (classification) and equivariant (crop prediction, color prediction) tasks in CIFAR-10 under the environment of crop, i.e. CONTEXTSSL (crop), and color, i.e. CONTEXTSSL (color).

| Method | Crop prediction ($R^2$) | | | | | Color prediction ($R^2$) | | | | | Classification (top-1) |
|---|---|---|---|---|---|---|---|---|---|---|---|
| | 0 | 2 | 14 | 30 | 126 | 0 | 2 | 14 | 30 | 126 | Representation |
| SimCLR | | | 0.459 | | | | | 0.154 | | | 89.1 |
| SimCLR$^+$ (c=0) | | | 0.448 | | | | | 0.054 | | | 88.9 |
| SimCLR$^+$ | | | 0.362 | | | | | 0.318 | | | 59.9 |
| CONTEXTSSL (crop) | 0.606 | 0.606 | 0.607 | 0.607 | 0.607 | 0.522 | 0.378 | 0.253 | 0.264 | 0.301 | 87.5 |
| CONTEXTSSL (color) | 0.605 | 0.467 | 0.387 | 0.466 | 0.511 | 0.523 | 0.525 | 0.527 | 0.527 | 0.527 | 87.5 |

Table 13: **CIFAR-10 Crop-Blur.** Performance of CONTEXTSSL on equivariant (crop prediction, blur prediction) tasks in CIFAR-10 under the environment of crop, i.e. CONTEXTSSL (crop), and blur, i.e. CONTEXTSSL (blur), to predict individual latent values.

| Method | Crop prediction ($R^2$) | | | | | Blur prediction ($R^2$) | | | | |
|---|---|---|---|---|---|---|---|---|---|---|
| | 0 | 2 | 14 | 30 | 126 | 0 | 2 | 14 | 30 | 126 |
| SimCLR | | | 0.382 | | | | | 0.122 | | |
| SimCLR$^+$ (c=0) | | | 0.375 | | | | | 0.111 | | |
| SimCLR$^+$ | | | 0.202 | | | | | 0.322 | | |
| CONTEXTSSL (crop) | 0.576 | 0.575 | 0.576 | 0.576 | 0.576 | 0.835 | 0.795 | 0.630 | 0.644 | 0.663 |
| CONTEXTSSL (blur) | 0.575 | 0.504 | 0.463 | 0.443 | 0.474 | 0.835 | 0.835 | 0.836 | 0.837 | 0.837 |

Table 14: **CIFAR-10 Color-Blur.** Performance of CONTEXTSSL on equivariant (color prediction, blur prediction) tasks in CIFAR-10 under the environment of color, i.e. CONTEXTSSL (color), and blur, i.e. CONTEXTSSL (blur), to predict individual latent values.

| Method | Color prediction ($R^2$) | | | | | Blur prediction ($R^2$) | | | | |
|---|---|---|---|---|---|---|---|---|---|---|
| | 0 | 2 | 14 | 30 | 126 | 0 | 2 | 14 | 30 | 126 |
| SimCLR | | | 0.121 | | | | | 0.122 | | |
| SimCLR$^+$ (c=0) | | | 0.039 | | | | | 0.111 | | |
| SimCLR$^+$ | | | 0.242 | | | | | 0.322 | | |
| CONTEXTSSL (color) | 0.488 | 0.488 | 0.488 | 0.488 | 0.488 | 0.837 | 0.711 | 0.628 | 0.672 | 0.730 |
| CONTEXTSSL (blur) | 0.488 | 0.376 | 0.286 | 0.309 | 0.362 | 0.837 | 0.838 | 0.838 | 0.838 | 0.837 |

Table 15: **CIFAR-10 Crop-Blur.** Performance of CONTEXTSSL on equivariant (crop prediction, color prediction) tasks in CIFAR-10 under the environment of crop, i.e. CONTEXTSSL (crop), and color, i.e. CONTEXTSSL (color), to predict individual latent values.

| Method | Crop prediction ($R^2$) | | | | | Color prediction ($R^2$) | | | | |
|---|---|---|---|---|---|---|---|---|---|---|
| | 0 | 2 | 14 | 30 | 126 | 0 | 2 | 14 | 30 | 126 |
| SimCLR | | | 0.382 | | | | | 0.121 | | |
| SimCLR$^+$ (c=0) | | | 0.375 | | | | | 0.039 | | |
| SimCLR$^+$ | | | 0.202 | | | | | 0.242 | | |
| CONTEXTSSL (crop) | 0.570 | 0.572 | 0.572 | 0.572 | 0.572 | 0.495 | 0.417 | 0.342 | 0.356 | 0.373 |
| CONTEXTSSL (color) | 0.570 | 0.490 | 0.447 | 0.492 | 0.515 | 0.495 | 0.496 | 0.497 | 0.497 | 0.497 |

Table 16: **Context-Dependent Labels Classification Task.** Performance of CONTEXTSSL on equivariant (rotation prediction, color prediction) tasks for context-dependent labels to predict individual latent values. As context length increases, CONTEXTSSL becomes more equivariant to color (or rotation) and more invariant to rotation (or color) within the respective environment.

| Method | Rotation prediction ($R^2$) | | | | | Color prediction ($R^2$) | | | | |
|---|---|---|---|---|---|---|---|---|---|---|
| | 0 | 2 | 14 | 30 | 126 | 0 | 2 | 14 | 30 | 126 |
| SimCLR | | | 0.781 | | | | | 0.058 | | |
| SimCLR$^+$ (c=0) | | | 0.478 | | | | | -0.003 | | |
| SimCLR$^+$ | | | 0.695 | | | | | 0.267 | | |
| CONTEXTSSL (color) | 0.751 | 0.751 | 0.750 | 0.750 | 0.749 | 0.915 | 0.973 | 0.980 | 0.981 | 0.981 |
| CONTEXTSSL (rotation) | 0.750 | 0.778 | 0.797 | 0.795 | 0.795 | 0.915 | 0.375 | 0.104 | 0.091 | 0.090 |

(rotation prediction) task in Table 17 and Table 18. CON-TEXTSSL maintains almost the same performance for rotation prediction using either representations or embeddings, while the performance of all other baselines drops significantly when using the embeddings. Similar conclusions apply to the classification case, except for SimCLR$^+$, for which the classification accuracy for both representations and embeddings is low.

### C.7. Enforcing Invariance or Equivariance to the Same Transformation Using Context

Apart from adaptively learning equivariance to a subset of transformation groups and invariance to the rest as shown in Table 1, we extend CONTEXTSSL to operate within environments characterized by a single transformation. Motivated by this, we ask the question: *Can* CONTEXTSSL *adapt to learn equivariance or invariance to the same transformation depending on the context?*. At training, we randomly sample one of these environments. If the environment corresponds to enforcing equivariance, we construct our context in the same way as before i.e. pairs of positives transformed using augmentations sampled from the transformation group. However, if the environment corresponds to enforcing invariance, we maximize alignment between positives transformed by augmentation sampled from the transformation group without conditioning on that augmentation. Take rotation in 3DIEBench as an example. As shown in Table 19, similar to our results in two transformation setting (rotation and color) in Table 1, CONTEXTSSL effectively adapts to enforce invariance and equivarance to rotation depending on the context. Results for predicting individual latents are provided in Table 20.

Table 17: Model performance in rotation prediction task, within the rotation-equivariant environment. The $R^2$ values are calculated for both the representations and the embeddings (output of projection head for invariant models (VICReg, SimCLR) or predictor for equivariant models (SEN, EquiMod, SIE, CONTEXTSSL). Unlike other models, which experience a significant performance drop between representations and embeddings, CONTEXTSSL maintains consistent performance.

| Method | Rotation prediction ($R^2$) | | |
|---|---|---|---|
| | Representations | Embeddings | Change |
| VICReg | 0.37 | 0.23 | -0.14 |
| SimCLR | 0.51 | 0.23 | -0.28 |
| SEN | 0.63 | 0.39 | -0.24 |
| EquiMod | 0.51 | 0.39 | -0.12 |
| SIE | 0.67 | 0.60 | -0.07 |
| CONTEXTSSL (rotation) | **0.74** | **0.74** | **-0.00** |

Table 18: Performance of CONTEXTSSL on accuracy of predictor embeddings for context-dependent labels.

| Method | Classification (top-1) | | | | | | |
|---|---|---|---|---|---|---|---|
| | 0 | 2 | 14 | 30 | 126 | Representation | Change |
| SimCLR | | | 52.7 | | | 85.3 | -32.6 |
| SimCLR$^+$ (c=0) | | | 72.4 | | | 83.4 | -11.0 |
| SimCLR$^+$ | | | 41.8 | | | 42.3 | -0.5 |
| CONTEXTSSL (rotation) | 76.6 | 76.9 | 75.6 | 76.9 | 77.5 | 80.4 | -2.9 |
| CONTEXTSSL (color) | 76.6 | 75.3 | 71.7 | 72.6 | 76.5 | 80.4 | -3.9 |

Table 19: **Single Transformation Setting.** Performance of CONTEXTSSL in 3DIEBench under the equivariant environment, i.e. CONTEXTSSL (rotation), and the invariant environment, i.e. CONTEXTSSL (none), with respect to rotation.

| Method | Rotation prediction ($R^2$) | | | | | Classification (top-1) |
|---|---|---|---|---|---|---|
| | 0 | 2 | 14 | 30 | 126 | Representation |
| SimCLR | | | 0.506 | | | 85.3 |
| SimCLR$^+$ (c=0) | | | 0.478 | | | 83.4 |
| SimCLR$^+$ | | | 0.247 | | | 42.3 |
| CONTEXTSSL (rotation) | 0.737 | 0.737 | 0.736 | 0.737 | 0.738 | 80.6 |
| CONTEXTSSL (none) | 0.737 | 0.717 | 0.477 | 0.377 | 0.473 | 80.6 |

Table 20: **Single Transformation Setting.** Performance of CONTEXTSSL in 3DIEBench under the equivariant environment, i.e. CONTEXTSSL (rotation), and the invariant environment, i.e. CONTEXTSSL (none), with respect to rotation, to predict the individual latent values.

| Method | Rotation prediction ($R^2$) | | | | |
|---|---|---|---|---|---|
| | 0 | 2 | 14 | 30 | 126 |
| SimCLR | | | 0.791 | | |
| SimCLR$^+$ (c=0) | | | 0.773 | | |
| SimCLR$^+$ | | | 0.544 | | |
| CONTEXTSSL (rotation) | 0.778 | 0.777 | 0.767 | 0.768 | 0.777 |
| CONTEXTSSL (none) | 0.839 | 0.829 | 0.721 | 0.667 | 0.698 |

