# OpenReview forum: "In-Context Symmetries: Self-Supervised Learning through Contextual World Models"
_ICML.cc/2024/Workshop/ICL — ICML 2024 Workshop ICL Poster_

### Official Review · Reviewer_nmCK · 2024-06-06
**Novel mechanistic approach to in-context self supervised learning**

**Rating:** 3
**Fit:** 3
**Confidence:** 2

**Workshop Review:**

Most self supervised learning methods explicitly bake in invariances or equivariances inductive biases. On the other hand humans are able to infer and represent such symmetries on the fly.  Such adaptability remains elusive to modern neural networks and understanding the underpinnings of adaptibility is central to human cognition. To this end, the paper proposes a novel approach to this problem.

A GPT like transformer assimilates the context and predicts the encoded representations that are then optimized with the usual contrastive learning losses. Naively using a transformer runs the risk of shortcut learning, memorization and invariance collapse. This work circumvents these issues through masking and an auxiliary prediction task.

Overall, the paper is well written and motivated and the approach clearly articulated. The experiments are fleshed out well, exploring challenging datasets that setup confounding invariances and equivariances. Some constructive feedback:

(1) Page 5: "a trivial solutionn to minimizing the alignment loss arises where the model treats the embeddings of (x_i,a_i) identical to y_i." this statement sounded confusing on a first read. Authors should try disambiguating this.

(2) For the N^+ version of the baselines: traditional self supervised methods are context-agnostic, how did the authors incorporate the context?  A pictorial depiction of this would be helpful to understand the modification.

(3) Table 1: Why are baselines only run on a specific context length while performing the full sweep for ContextSSL?  Comparing the sensitivity of Context SSL to context length vs baselines is important to establish.

(4) Fig. 4: "while the model retains its performance on the invariant classification task, it fails to learn equivariance ,and cannot effectively adapt to different contexts." to the best of my understanding fig. 4 only shows equivaraince prediction results. the invariance results seem to be missing.

(5) Appendix C.1.2 : Missing references

(6) Existing SSL methods enjoy their status quo because traditional data augmentations are cheap and straightforward to obtain. In the light of this, a line or two about hwo the authors think their method can be scaled to in-the-wild images would be beneficial for the community.

Despite of some drawbacks, all-in-all this paper forges an important path forward into in-context learning for visual stimuli.

**Reason For Not Giving Higher Score:**

Experiments could be more detailed and extended to natural stimuli.

**Reason For Not Giving Lower Score:**

N/A

---

### Official Review · Reviewer_EMeM · 2024-06-08
**Do data-augmentations as ICL, so that the model can pick invariance or equivariance contextually.**

**Rating:** 2
**Fit:** 3
**Confidence:** 2

**Workshop Review:**

This paper attempts to tackle the rigidity of data-augmentations in self-supervised learning. The authors argue that the pre-defined set of augmentations might not be suitable for downstream tasks due to their rigidity and instead proposes a new architecture to move data-augmentations as context. The context contains demonstrations that inform the model about the task-specific symmetries. This is achieved by modifying the Image World Models (IWM) architecture to include context.

In addition to creating a new architecture, the authors had also to devise masking strategies so that the the model will not take shortcuts or learn everything as invariance. These insights could be useful for other models too. However, these additional mechanisms increase the complexity of the training pipeline and raises questions about the generality of it.

The datasets used are adequate to evaluate the idea and the paper includes a number of ablations that provide insight on the internal mechanisms.

However none of the experiments rise to the level of showing the language-model-like ICL abilities. What are shown seem to be pure conditioning effects -- invariances and equivariances are both learned for every transformation, and the prompt just conditions the model to select which one.

The authors should think about a test, perhaps using combination of transformations with multiple objects to demonstrate LM-like ICL.

Overall this is an interesting approach and worth inclusion in the workshop for discussion.

**Reason For Not Giving Higher Score:**

However none of the experiments rise to the level of showing the language-model-like ICL abilities. What are shown seem to be pure conditioning effects -- invariances and equivariances are both learned for every transformation, and the prompt just conditions the model to select which one.

The authors should think about a test, perhaps using combination of transformations with multiple objects to demonstrate LM-like ICL.

**Reason For Not Giving Lower Score:**

The paper is well written (though a bit repetitive at times), has nice justifications for the brittleness of data-augmentations in SSL, and has ablation studies to show the properties of the architecture.

---

### Meta-Review · Area_Chair_51vx · 2024-06-14

**Recommendation:** 2

**Metareview:**

This papers presents a contrastive learning framework that uses a transformer module to adapt to selective invariance or equivariance to transformations by focusing on task-related context. Authors show that learning with context is prone to identifying shortcuts and subsequently propose masking and auxiliary prediction task. Extensive results and ablations are shown.

All reviewers consider this paper favorably.

---

### Decision · Program_Chairs · 2024-06-17

Accept (Poster)